# Development of a Swine Production Cost Calculation Model

**DOI:** 10.3390/ani12172229

**Published:** 2022-08-30

**Authors:** Laya Kannan Silva Alves, Augusto Hauber Gameiro, Allan Paul Schinckel, Cesar Augusto Pospissil Garbossa

**Affiliations:** 1Laboratory of Swine Research, Department of Animal Nutrition and Production, School of Veterinary Medicine and Animal Science, University of São Paulo, Pirassununga 13635-900, Brazil; 2Laboratory of Socioeconomic Analysis and Animal Science, Department of Animal Nutrition and Production, School of Veterinary Medicine and Animal Science, University of São Paulo, Pirassununga 13635-900, Brazil; 3Department of Animal Sciences, College of Agriculture, Purdue University, West Lafayette, IN 47907, USA

**Keywords:** agribusiness, expense, management, modelling, pig production

## Abstract

**Simple Summary:**

Swine production is a for-profit activity; however, most farms have deficient internal controls and empirical management and do not even know the cost of the market hog produced. Knowing the production cost of what will be commercialized is crucial for any process that involves business management, and in pig production, it is no different. However, the lack of a standard method and simple and easily accessible tools make it difficult for producers to organize the economic management of their businesses. In this sense, the present work aimed to develop a free and easy-to-use tool that calculates swine production costs and serves as a management tool in commercial properties.

**Abstract:**

This paper aims to present a tool that offers pig producers a standard method to calculate and control their production costs and, consequently, provides the necessary information to guide strategic decision-making. Following these premises, a mathematical model to estimate swine production costs were developed using Microsoft Excel^®^ software (version 2207). Case studies were used to assist in the characterization and construction of the model. Through the panel method, the tool was validated by professionals in the sector. Costs were considered according to the Neoclassical Economic Theory of Costs and allocated in the order of variable costs, fixed operating costs, and opportunity costs of capital and land. These costs together create the total cost. The model provides the total cost per batch, per market pig, per arroba, and per kilogram, which facilitates the interpretation of the results and economic evaluations of the system. The model is adaptable to different types of swine farming, as well as the consideration of all costs involved in the production system, whether explicit or implicit. The model developed has the potential to be used as a management tool in commercial swine production systems, assisting the producer in the decision-making process through the management and control of production costs.

## 1. Introduction

Commercial pig production is marked by the continuous search for productivity and profitability [1,2], which takes place through the efficient use of production factors. In this sense, like any for-profit organization, pig production requires dynamic management [3]. Thus, it is necessary to economically analyze the activity, which makes it possible to use economically and profitably land, labor, and capital.

Through economic and productive analysis within the farm, the manager becomes aware of the results obtained in the activity, able to verify the production process step by step and thus locate the bottlenecks within the management of his business [1,2,3]. However, one of the major difficulties found in Brazilian pig farming is that often the farms have deficient internal controls and are managed empirically, not being able to obtain the information that should guide the decision-making process: the cost of the pig to be commercialized.

With a market evidenced by dependence on commodity prices [4] and volatility in the sale price, swine production requires that the producer has absolute control over his production [5], allowing for good results to appear. Pig farming requires rational choices regarding the efficient use of production factors since the choices made in carrying out the activity impact the total cost, which in turn will directly impact the economic results [6]. Thus, it emphasizes the importance of knowledge, management, and control of production costs to maximize results and support strategic decision-making [7,8].

The lack of a previously defined standard protocol for the calculation of production costs is one of the main factors that make it difficult to carry out cost analysis in livestock activities [9]. In the literature, there are some models for calculating pig production costs in line with Brazilian pig production [10]. However, the available tools are aimed at the management of integrated farms and do not consider the costs of feeding the animals, which can represent between 65% and 80% of the production costs of pigs in Brazil [11].

The objective of this study was to develop a mathematical model to estimate swine production costs that would be flexible to meet the most varied types and organizational arrangements of swine production systems existing worldwide and offers pig producers a standard method to calculate and control their production costs, and consequently, provides the necessary information to guide strategic decision-making.

## 2. Materials and Methods

Research on animals was conducted according to the institutional ethics committee on animal use (CEUA nº4257240619) and the institutional research ethics committee from Brazil Platform (CEP 3.780.942).

This research was carried out in Pirassununga, State of São Paulo, Brazil, and its development consisted of three steps. Firstly, case studies were carried out to provide a detailed description of the production process and the characteristics of the independent swine production systems. In the second stage, there was the development of the model for calculating the costs of pig production, based on the Neoclassical Economic Theory of Costs and the characteristics of the production systems defined in the first stage. In the third stage, the presentation and validation of the calculation model for the agents in the swine production chain were carried out.

### 2.1. Characterization of Swine Production Systems

The first stage of the study consisted of carrying out case studies to understand in detail the production process and the characteristics of the complete cycle swine production systems. The case study investigates methodology and addressed questions that refer to “how” and “why” the research is being conducted. The questions of the study were formulated to define the swine production unit to be analyzed, following the procedures proposed by Yin [12]. The conditions were as follows: (a) that it should be a representative unit of the activity that it was intended to model; (b) that it would be possible to visit the property as many times as necessary for proper data collection and construction of the model; (c) that farms with different sizes and levels of technological adoption were evaluated. The item “c” is very important since the productive characteristics are somewhat heterogeneous for the swine-producing farms, mainly in terms of physical space, the housing of sows, herd size, levels of automation, forms of management, and coexisting activities on the property, among others.

After identifying pig producer partners, two case studies were conducted. Technical visits were carried out on two farms, and a questionnaire was given to the property manager as a way of validating the data collected with production employees, sector leaders, and technical managers.

Information regarding the management of the farms was collected, and main fees and taxes were paid by pig farming, machinery, equipment, vehicles, buildings, and facilities. Nutritional, reproductive, and sanitary management, health, and biosecurity programs adopted on the farms were identified and described. The zootechnical variables of importance to control the productive and reproductive performance of each animal in the systems were listed. A survey was also carried out of the labor present on the site of each of the collaborating farms.

The researchers also developed a SWOT analysis of Brazilian pig production to assess the activity. The SWOT analysis is a classical strategic planning instrument [13] that provides a simple way to estimate the best form to implement a strategy [14]. In using strengths and weaknesses and external opportunities and threats as the framework this analysis helps the researchers gain insight into what is usually achieved and what things should be considered when developing the cost calculation tool. In this sense, the SWOT analysis was made according to Houben et al. [15].

### 2.2. Allocation of Production Costs and Development of the Calculation Model

Microsoft Excel^®^ software (Microsoft 365 MSO, version 2207–64 bits, Redmond, WA, USA) was the tool used to develop the calculation model, in which all costs of the full-cycle swine production activity were included. All data collected in the field were considered and allocated as proposed by the Neoclassical Economic Theory of Costs, which guided the logic of costs, as Raineri et al. [7] proposed for sheep production and Sartorello et al. [16] for feedlot beef cattle. Adaptations were also carried out based on the proposed cost method for agriculture, such as that of the National Supply Company (Conab) [6]. The objective was to meet the theoretical precepts to facilitate the practical use of the model, following a logic that is easy to understand and adopt for pig farmers. The cost components were allocated in the following order: (i) variable costs; (ii) fixed operating costs; (iii) operating costs; (iv) capital and land remuneration costs; and (v) total cost.

In variable costs, all the components that only change according to the quantity produced were grouped. Fixed costs, on the other hand, do not change when the volume of production varies in the short term. In the item “Compensation costs for capital and land”, the remuneration of the land and capital represented the opportunity cost of these production factors. Opportunity cost is a theoretical concept that represents the benefits that could have been obtained from an unchosen opportunity [7,8,16]. The total cost was obtained through the sum of variable costs, fixed costs, and costs of remuneration of capital and land.

### 2.3. Model Validation

After structuring the calculation model, the information collected in the case studies was gathered and organized in the Microsoft Excel^®^ model to generate the results proposed in the objectives, which consisted of the development of the mathematical model for calculating the production costs of pigs in a complete cycle and availability of the tool to the public. Later, using the Panel method, adapted from Raineri et al. [7], the calculation model developed was validated with swine producers in the state of São Paulo.

The Panel method included a meeting between researchers and a group of producers and technicians from the sector studied, during which participants discussed the model in detail and made the relevant changes [7]. Thus, all calculation formulas and results found in the mathematical model were presented in detail to producers and technicians in the São Paulo swine sector.

A Term of Commitment, Secrecy, and Confidentiality was signed between researchers and those responsible for the enterprises in order to conduct the research. This term clarifies that all technical information obtained would be used solely and exclusively for the development of the research and would be considered confidential, as well as the identity of the property and owner.

## 3. Results

A SWOT analysis was developed to depict the state of Brazilian pig production (Figure 1). This is a strategic planning tool used to assess the activity, focusing on the strengths and weaknesses as well as the opportunities and threats of producing a market hog.

The proposed cost allocation scheme aligns with the Neoclassical Economic Theory of Costs (Table 1).

According to the presented cost allocation, the item organization was divided into five groups: (A) variable costs; (B) fixed operating costs; (C) operating cost; (D) cost of remuneration for capital and land; and (C) total cost. The sum of items A and B results in item C, and the sum of items C and D results in the total cost of the activity, as described in Equations (1) and (2):(1)$$COP_{l}=CV_{l}+CFOP_{l}$$
(2)$$CT_{l}=COP_{l}+CO_{l}$$

In Equation (1), *COP_l_* represents the operating costs of swine production, referring to the weekly batch *l*, in Brazilian currency (the “Real”). *CV_l_* represents the variable costs of swine production, referring to the weekly batch of animals *l*, in Brazilian real. *CFOP_l_* represents the fixed operating costs of swine production, referring to the production of the batch of animals *l*, in Brazilian real. In Equation (2), *CT_l_* represents the total cost per batch of swine produced, referring to the weekly batch *l*, in Brazilian real; and *CO_l_* represents the weekly cost of remuneration for capital and land, in Brazilian real. It is important to note that the cost report is given based on the weekly batch *l* to be marketed. However, the costs related to the entire production cycle that led to the production of this batch are computed.

In full-cycle swine farms, what dictates the flow of production, operation, and the inner workings of the activity are the female groups, which are usually weekly groups. A weekly group of females is nothing more than a set of females that will be inseminated during the week to meet the demand, farrowing and weaning schedule, and the productive cycle of the farm. Thus, it was decided to carry out the calculations of consumption of inputs used in the production of swine weekly. The following formula was used to calculate the weekly groups:(3)$$GS=\frac{N\times ppa\times txparto}{\varphi }$$

On what:

*GS:* females’ weekly group.

*N:* total number of dams in production on the farm.

*ppa*: number of births per sow per year on the farm.

*txparto:* farm’s birth average rate.

*φ:* number of weeks contained in a 365-day year.

The presentation of mathematical equations and nomenclatures used in the construction of the cost calculation model will be presented according to the allocation to facilitate understanding. Thus, the costs of all inputs are allocated as variable costs (Table 2), and the nomenclatures and definitions of the formulas used (Table 3) are presented next.

Variable costs (*CV_l_*) of the activity were calculated from the input costs (*CCA; CS; CR; CB; Cta; Cs; CTdfv*) of each item consumed by the animal categories c involved in the production system of the different production phases f present in a commercial swine farm in full cycle. These inputs, in turn, are consumed to produce the batch of animals *l* to be marketed weekly. In the animal categories, gilts, dams, ruffians, boars, piglets, and market hogs are examples. Regarding the phases, one can mention the sow in the gestation phase, the sow in the lactation phase, a piglet in the farrowing room, and/or a piglet in the nursery phase, a growing and/or finishing pig, for example.

The costs computed as costs with feeding the productive herd refers to using different diets with controllable periodicity. To adjust the model and better estimate costs, the period of supply of each of the diets related to an animal category, as well as the amount supplied per day, in kilograms, of this same diet are to be entered. The product of the period and quantity supplied results in feed consumption per animal in the phase, as shown in Equation (6). These data are then crossed with data from the productive herd, previously informed, and thus, the weekly feed intake of the phase is obtained (Equation (5)), including all animals receiving that diet in the specified phase. The weekly feed consumption in phase is multiplied by the price of the specified diet, and the cost of feeding animal category c in phase f is found, as described in Equation (4). Subsequently, the sum of the weekly feed costs of all categories and phases present in the production system is carried out, and the subtotal of the costs of feeding the group of animals is obtained, as can be seen in Equation (7).

The costs computed as health costs refer to the use of vaccines and medicines with a controllable frequency in the herd. In the case studies, different items were found in each of the units studied, thus enabling the adequacy within the model to include the input to be used and the number of doses used to treat the animals per phase for a given animal category. The product of the number of doses per phase and the number of animals in the phase results in the calculated quantity of doses of vaccines and medicines used weekly for a certain animal category in that phase (Equation (9)). By crossing these data with the price of the dose used, the cost with the health of the animal category c in phase f is obtained, as shown in Equation (8). Subsequently, the sum of the weekly costs with the veterinarian’s expenses is performed in all categories and phases present in the production system, and the subtotal of weekly costs with the health of the productive herd is obtained, described in Equation (10).

The costs computed with reproductive management were divided into three possible scenarios, namely: (i) the production of semen on the property, where the costs of semen collection, processing supplies, and laboratory equipment for semen evaluation are considered; (ii) the use of semen from a central distributer, where the quantity and prices of the commercial semen dose to be used in the property by gilts and sows are considered; and (iii) where the two scenarios are used, both internal semen production and the use of semen from a central distributer, and all these inputs are taken into account. In the three scenarios described, the materials used in the artificial insemination procedure, such as pipettes and tubes, are also considered. Through the weekly group of females and data previously entered in the model, it is possible to identify the number of inputs related to product management to be consumed weekly by the system (Equation (12)). The product of the weekly amount of inputs used and the unit price of such inputs result in the cost of reproductive management for the farm, as shown in Equation (11). The sum of costs per week for each scenario allows the weekly subtotal cost to be obtained with reproductive management on the farm (Equation (13)).

The model was adapted to allow the insertion of the item I to calculate expenses with consumer goods, as well as the average monthly quantity consumed of this item and the unit price of each item entered. Subsequently, to represent the goods consumed by the weekly batch *l*, the value found was divided by 4.345 (ω), which represents the number of weeks present in a month (Equation (14)). Performing the sum of expenses related to consumer goods such as gloves, needles, syringes, marker sticks, and control of rodents and flies, among others, the subtotal of expenses with consumer goods for the farm is then obtained (Equation (15)).

These subtotals made up what was called the “Pig Raising Subtotal Defrayal”, which is nothing more than the sum of the variable costs of food, health, reproduction, and consumer goods (Equation (16)). This data was characterized as important by the producers and managers of the farms where the case studies were carried out, as it allows the visualization of costs related directly to the animals.

Subsequently, transportation, loading, and insurance expenses for lot l to be sold were calculated. The data referring to the distance, in kilometers, from the farm to the slaughterhouse and the price paid to the carrier per kilometer traveled were imputed (Equation (17)) to calculate the cost of transport. The rate charged by the insurer and the average capital invested in the lot is considered to calculate the lot insurance, as described in Equations (18) and (19). The sum of both results in the subtotal of weekly costs with transport, loading, and insurance of the weekly lot to be sold (Equation (20)).

The last item that makes up the variable costs refers to the weekly variable financial expenses. This item, consisting of variable taxes and rates such as ICMS, Funrural, and GTA, among others, considered the main expenses raised in the case studies and followed the current accounting regulations for the calculation. The details are found in Equation (21).

The equations for fixed operating costs and the nomenclatures and definitions of the formulas used are shown in Table 4 and Table 5.

In the composition of the fixed operating costs (*CFOP*_l_) of the activity, there are costs related to labor, electricity, telephone, internet, fuel, costs with depreciation, maintenance, and other fixed expenses, which include taxes and fixed fees collected from the activity. For the calculation of operational fixed items, it may eventually be necessary to apportion some items, as they are used in activities other than the swine production. In these cases, the user must allocate the data that represent the proportional costs referring to the use of the items by the swine production activity.

The amounts computed as labor refer to costs related to the salary payment of all employees involved in the swine production process. To adjust the model and better estimate, the user must impute the employee’s salary, with all charges and bonuses already included, in Brazilian currency per month. By crossing such data with the number of employees exercising a certain activity, the weekly cost related to labor is obtained, as shown in Equation (23). The sum of the weekly labor costs of all employees on the farm results in the subtotal of weekly labor costs (Equation (24)).

Only the consumption related to the swine production activity should be considered for the item electricity costs. For this, the number of KWh consumed monthly by swine farming and the rural tariff for electricity paid by the property, in R$/KWh, will be imputed. The multiplication of these two items provides the monthly cost of electricity by the activity; subsequently, this value is divided by ω, and the weekly cost of electricity is obtained (Equation (25)). For the calculation of weekly costs with telephone and internet services (Equation (26)), the prices of the packages used by the property were considered. In the case of more than one activity using this service, the cost must be apportioned, and only the portion referring to the swine production must be imputed in the model.

The amounts computed as fuel costs refer to those related to the consumption of gasoline, alcohol, and diesel by the swine production process. To carry out the calculation, the user must input the price data per liter of fuel used and the number of liters consumed monthly (Equation (27)). Subsequently, the model divides the sum of these items by the fixed value of the number of weeks present in a month, thus obtaining the weekly fuel cost (Equation (28)).

The item depreciation of assets corresponds to the financial reserve necessary to acquire an asset with the same characteristics at the end of its useful life. For the calculation of depreciation of swine production assets (Equation (29)), which includes improvements and facilities, machinery and equipment, and biological assets, the straight-line method was used, in which it is necessary to estimate the useful life in years, and the residual value of the production good. In practice, the residual value of the asset is known as a percentage of the new asset, as shown in Equation (31). The unit price of fixed capital and the unit quantity of assets are considered to calculate the initial values of the new asset (Equation (30)). In the specific case of installations, the Pcj can be given by the Basic Unit Cost of Civil Construction or by the real value in each specific case. For other capital goods, it corresponds to their market value. With such information imputed in the model, it is possible to calculate the weekly costs with capital depreciation (Equation (32)).

The model also calculates the periodic maintenance of all items used in the swine production system. A rate established by the activity manager is multiplied by the value of the capital good in question to calculate the maintenance of the different capital goods used in the different productive sectors of the farm (Equation (33)). Subsequently, it is then possible to calculate the weekly costs with maintenance and conservation of all goods used by the activity (Equation (34)).

The last item that makes up the fixed operating costs refers to the weekly fixed financial expenses. This item, consisting of taxes and fixed rates such as ITR, associations, and union memberships, among others, considered the main expenses raised in the case studies and followed the current accounting regulations for the calculation. Details can be found in Equation (35). With all this information imputed, it is then possible to calculate what is called the operational cost of the activity, as shown previously in Equation (1).

Finally, the remuneration cost of capital and land, or the opportunity cost of capital and land, was divided into three items and sub-items (Table 1 in item D) to facilitate understanding and can be calculated by Equation (37).
(37)$$CO_{l}=RCI_{l}+RCG_{l}+RCT_{l}$$
where:

$RCI_{l}$: Remuneration cost on fixed capital.

$RCG_{l}$: Cost of remuneration on working capital.

$RCT_{l}$: Opportunity cost of land lease.

The opportunity costs of capital and land, in the proposed model, are remuneration for the use of equity capital. An interest rate must be used to remunerate working capital, capital invested in assets, and land use to be calculated. The details of the calculation methods for each of the items (Table 6), as well as the nomenclatures and definitions of the formulas used (Table 7), can be seen in the following section.

The remuneration of fixed capital (Equation (38)) by the producer refers to the portion calculated on the value of the property purchased and used in production and included in the fixed operational cost of production [6], such as facilities, equipment, dams, and gilts for replacement. The rate used to remunerate fixed capital is defined by the activity manager in advance in the system’s characterization data.

The remuneration of working capital, shown in Equation (39), refers to the capital used to fund the raising of the animals—as shown above (Equation (16))—present in variable costs. The rate used to remunerate working capital is also the manager’s option and must be previously entered into the system’s characterization data. An example of a rate to be used, both for the *RCG* and for the *RCI*, is the Reference Rate of the Special Settlement and Custody System (Selic). The period used as the turning period was weekly to monitor the production costs of a weekly batch *l*.

For remuneration of the land used by the swine production activity, the method that relates land use with the land lease value in the region where the property is located was chosen and can be observed in Equation (40). For this, it is necessary to know and impute the lease value in the characterization tab of the production system.

After calculating the variable costs, fixed operating, and opportunity costs, it is then possible to obtain the total cost of production of the swine. As shown in Equation (2), the total cost is nothing more than the sum of these costs. In the calculation model, this total cost can be observed in a total monetary amount, or in total cost per market pig, per arroba, and per kilogram of the swine produced. The equations (Table 8) that demonstrate the calculation method to obtain such indicators are presented below in Table 8, as well as the acronyms and nomenclatures used (Table 9).

To carry out such calculations as specified in Equations (41)–(43), it is necessary to know the number of animals finished at the end of the accommodation of lot *l* (*Nt_l_*). The number of animals housed and the mortality rate of each phase was considered in each of the phases, as shown in Equation (44).
(44)$$Nt_{l}=\underset{cf}{\sum }Na_{cf}\left(1-Txmort_{cf}\right)$$
where:

$Na_{cf}$: is the number of animals destined for slaughter, in the phase *f*.

$Txmort_{cf}$: is the mortality rate reported by the user, for a given phase *f*.

The base number of animals in each of the phases was established based on the number of animals born alive, which is obtained as shown in Equation (45).
(45)$$Na_{cf}=GS\times Nnv\;\left(1-Txmort_{cf}\right)$$
where:

$Na_{cf}$: is the number of animals in the farrowing room.

Nnv: is the number of piglets born alive.

$Txmort_{cf}$: the pre-weaning mortality rate of piglets.

To obtain the production in kilograms of market pigs, the Equation (46) was used.
(46)$$\theta_{l}=Nt_{l}\times Pcab_{l}$$
where:

$\theta_{l}$: represents the quantity, in kilograms produced at the end of batch *l*.

$Pcab_{l}$: is the average weight, in kilograms, per finished pig at the end of the batch *l*.

The number of weekly kilos to be sold was also calculated from the culling animals of the reproductive herd, both males and females. For such calculations, the number of animals discarded weekly and the average weight of these animals were considered (Equation (47)).
(47)$$\vartheta_{l}=\frac{Na_{cf}\times txdes_{c}}{\varphi }\times Pdes_{c}$$
where:

$\vartheta_{l}$: represents the quantity, in kilograms, produced weekly in the detriment of the disposal of s sex animals sold together with the batch *l*.

$Na_{cf}$: number of animals of the sex *s* present on the farm.

$txdes_{c}$: is the annual culling rate, of males or females, expressed as a percentage.

$Pdes_{c\;}$: cull weight of animal on category *c*.

The model also calculates the profitability of the swine production system, which is shown in the economic report. The indicators analyzed are (i) total weekly revenue, (ii) profit or loss per weekly batch, (iii) total income to the producer, (iv) leveling point, (v) total factor productivity, and (vi) return on investment. The calculation formulas for these economic indicators (Table 10), as well as the nomenclatures and definitions listed in the equations (Table 11), are presented next.

Both the revenue from the sale of market pigs and the revenue from the sale of culling animals were considered to calculate the total weekly revenue from the swine production activity. The prices used to calculate the revenue are previously informed by the user in the system characterization tab, where the market value of the swine sold in the farm region in the period of marketing of the batch of animals *l* is used, as already demonstrated in Equation (48). After calculating the total weekly revenue, it was then possible to calculate the average revenue per market pig produced and per kilogram of market pig produced, shown in Equations (49) and (50), respectively. With the results of the total weekly revenue, it is then possible to obtain the economic profit of the activity or loss. The calculation of profit is nothing more than the subtraction of the total cost of production of the batch of animals *l* from the total weekly income (Equation (51)), swine producers consider this one of the main economic indicators of the system.

The Leveling Point (Equation (52)) was also calculated, which is an economic indicator that demonstrates the level of production at which the value of marketing the product equals the costs to produce an item, in this case, the head or market pig produced. In other words, this indicator estimates the minimum quantity to be produced in this system with current productivity data, with these costs, so that there is no economic loss. The Benefit Cost Ratio, calculated as shown in Equation (53), is an indicator that shows revenue per cost unit. The *RBC* then demonstrates how much the swine farmer is receiving for the production activity for each Brazilian real spent on the production of the batch under analysis. Another economic indicator analyzed is the Return on Investment (Equation (54)). Through this indicator, it is possible to know how much the swine production activity is gaining (or losing) concerning the investment made. This indicator is shown in percentage.

Finally, there is the calculation of the producer’s total income, which considers the profit of the activity, the remuneration for the work of the producer and his family, which is nothing more than the wage compensation for corporate officers, and the remuneration on capital and the earth (Equation (55)). In this case, the factors of production are considered specific to the swine producer. Thus, the opportunity cost of capital and land, which is a cost of the activity, becomes the revenue for the producer since it is composed of the remuneration of the equity invested in pig farming. The producer’s total income is given in monetary amounts and in reais per kilogram of market pig produced.

In a panel held with the São Paulo Association of Swine Producers, its representatives, and associates, the developed calculation model was evaluated, validated, and considered as complete, adequate, and representative of the swine production systems, fulfilling the proposed objectives.

## 4. Discussion

Through SWOT analysis, it was possible to identify both internal factors (which can be improved on-farm) and external factors that may affect pig production. When it comes to for-profit activities, one of the greatest challenges is related to producing efficiently, i.e., at lower costs, and how to run off the production [5]. In this sense, attention is drawn to two situations. Firstly, in pig production, the main raw materials in the feed composition are corn and soy [4,17]. Such items are commodities, and the price paid for these items cannot be controlled by the producer. Therefore, if the pig farmer cannot control the price of the feed base of his herd, which is equivalent to 65 to 80% of production costs [11,18], then it is necessary to manage all other expenses with the production of these animals. Additionally, to dependence on commodities, the producer also does not control the price of the market hog [4,19], which further strengthens the importance of cost management and control. Management is what makes the enterprise more viable, making it stronger in times of crisis, in addition to preparing to take advantage of opportunities [4].

The cost allocation scheme adopted was guided by some assumptions, aiming at objectivity to allow understanding, comparison, and decision making, without neglecting other effectively necessary items. The first of them was to develop a production cost calculation model that fits the Economic Theory [20]. The costs were grouped into variables, fixed operating and opportunity costs on capital and land, because it is believed that this allocation is easy for producers to understand. The segregated allocation of opportunity costs was made to make it clear to the producer that the opportunity cost is a cost of the swine activity. However, in the case of specific factors, it is a revenue for the producer. In this way, it is possible to visualize in the farm’s daily life what represents the cost of animal raising and items necessary to operate the activity, as well as what represents the remuneration of the pig farmer for the use of capital, whether immobilized or not. Such a cost allocation scheme was considered adequate and easy to understand by pig farmers in the state of São Paulo after panel validation, as well as corroborating the one adopted by Raineri et al. [7] and Sartorello et al. [16].

Variable costs include all components that participate in the process as the activity develops. That is, variable costs comprise all items that only affect production and are directly related to the number of swine produced [7,21]. As they are directly associated with the amount produced, variable costs are greatly influenced by zootechnical indicators, as these are responsible for dictating the level and flow of production on the farm. Feed, for example, is the item with the greatest impact on the composition of variable cost, as well as the total cost of swine production [18,22]. For this reason, the feeding efficiency within the farms is one of the most important indexes since that greater feed efficiency can directly impact total cost [23]. Therefore, for the use of the model as a management tool that aligns with the reality of the farm under analysis and contributes to strategic decision-making, it is extremely important to carry out consistent zootechnical bookkeeping. Zootechnical bookkeeping aims to raise indices that measure the efficiency of the production system, as well as point out the bottlenecks that may be affecting productivity and, consequently, the profitability of the system [24].

The labor was designated as a fixed cost because it is not directly associated with the number of animals housed or produced. It is known that there is a relationship between the number of animals housed per employee on a farm [25]. However, the number of employees or the number of hours worked cannot be increased by adding a unit produced since this increase is related to other factors such as the adopted production system, management carried out, and productivity, among others. In the literature, some authors agree [6,7,16] and diverge from this allocation, considering labor as a variable cost, as is the case of Girotto and Santos Filho [10], in calculating the cost of production of piglets, Miele et al. [26] in the calculation of the cost for broiler chickens and Santos Filho [27] for the cost of the swine producer for an integrated grow-finishing system.

Asset depreciation is a fixed cost item, corresponding to a financial reserve necessary to avoid the loss of capital of the swine producer. With such a reserve, it is possible, at the end of the asset’s useful life, to acquire another asset with the same characteristics [7,16]. Depreciation is nothing more than recognizing the loss of value of an asset to the detriment of natural wear and tear due to its use in the production process or its obsolescence. When it comes to depreciation, the use of the asset must always be considered in a temporal aspect. According to Silva et al. [28], the accounting of losses due to depreciation seeks to promote more efficient asset management, thus making it more reliable.

In practice, there is a certain difficulty for the producers in calculating the depreciation of their assets, and, according to Gameiro and Caixeta Filho [29], one of the reasons for such an impasse is to establish the ideal time for considering the depreciation of items such as fences, tractors, milking machines, feeders, and even the breeding herd. Currently, the Brazilian Federal Revenue Service provides some standards and practiced data that address the ideal useful lifetime for calculating the depreciation of assets [30]. However, the developed model is managerial and not accounting, not allowing the producer to follow the rules established by the tax authorities. Thus, it is possible to allocate the data to calculate the depreciation that best fits the reality of the farm under analysis.

This is one of the major bottlenecks found when analyzing studies that deal with the calculation of production costs. The difficulty in determining depreciation leads to studies that consider only variable costs, or only part of fixed costs, in the calculation, as in the case of the studies by Herrington and Tonsor [31] and Retallick et al. [32]. This is also true for maintenance, which is often left out in cost analyzes due to the difficulty in estimating a specific value [16]. In work by Santos et al. [33], for example, the depreciation of machines and implements are considered; however, there is no information about expenses related to maintenance. In work by Berthiaume et al. [34], both depreciation and maintenance of improvements and machines were not considered in the calculation. In practice, this can be a serious problem, as it underestimates the true value of the production cost of the activity.

For maintenance, some studies estimate the calculation of production costs of integrated livestock systems that allocate maintenance as variable costs [10,26,27,35,36,37]. However, in the present study, as well as in Raineri et al. [7] and Sartorello et al. [16], both depreciation and maintenance are computed, as well as allocated as fixed costs of the activity.

Another important item to be discussed is the opportunity cost of capital and land. According to Knight [38], all factors—land, labor, and capital—used to produce a given good must be remunerated, including opportunity costs, in which the cost of resources is equal to their value in the best alternative uses. This alternative use does not only represent the choice of the best internal alternative but also the value of using the most appropriate factor of production outside the company [8]. To facilitate the understanding of pig farmers in relation to the difference between activity cost and opportunity cost, it was decided to allocate the remuneration of the own production factors separately, as performed by Conab [6], Raineri et al. [7], and Sartorello et al. [16].

In the present study, in addition to considering the opportunity costs of immobilized capital and land, as commonly performed [6,33,39], and also the opportunity cost of working capital, which refers to the capital used to fund the raising, that is, spent on feed, veterinary supplies, reproductive management, among others, as shown above in Equations (16) and (38). Remuneration fixed capital, which in this study was subdivided into the items facilities, machinery and equipment, reproducers, and replacement gilts, refers to the portion calculated on the value acquired and used in production [6]. To remunerate both working and fixed capital, the interest rate levied on capital must be inserted in the model. However, this rate that remunerates the own production factors varies among researchers [16]. Therefore, it was decided in this study to keep the rate used as a criterion to be defined by the activity manager, who should use the rate that best represents the alternative use of capital used in funding and invested in the activity.

For the opportunity cost of land, a widely used criterion, which was adopted in this study, is the lease value in the region’s land market [4,7,16,33,40]. However, some authors diverge from this methodological consideration and use a fixed remuneration rate for land, as is the case of Conab [6], which rate at 3% per year.

The production cost is then calculated considering all costs, whether explicit or implicit, that occur in a swine production unit. It is important to include all the costs on the model for a realistic estimate of total cost and to be able to compare among the producers. According to Oliveira and Santos Filho [18], when segmented, the cost of production provides detailed information on the inputs and values consumed in the system, and from this information, it is possible to analyze the points of greatest sensitivity, thus making it possible to return efforts to plans of action and strategic decision-making for better use of production factors, cost reduction, and profit maximization. Also, management that considers cost strategy generates more agile decision-making, as it can objectively identify the loss [18].

Tools such as the model developed in the present study allow the producers more accurate management of their business, making it possible to monitor, simulate and predict risk scenarios for productive activity [25]. In this sense, the management and control of production costs increase farmer’s ability to keep in contact with each one of the inputs involved in the production and make informed daily management decisions. According to Vranken and Berckman [41], through observation, interpretation, and on-farm control, it is possible for the producer to achieve economically, environmentally, and socially suitable farming.

## 5. Conclusions

The developed model calculates swine production costs for different production systems, addressing all implicit and explicit costs involved in the production system. It has the potential to be used as a management tool, as well as to generate important information for decision-making in commercial swine production systems. The application of Neoclassical Economic Theory to calculate production costs, such as the opportunity cost, is essential for the development of more accurate models. These are the distinctive points of the cost calculation model elaborated in this work, which presents clarity and versatility for use by swine producers, technicians, and scientists.

## Figures and Tables

**Figure 1 animals-12-02229-f001:**
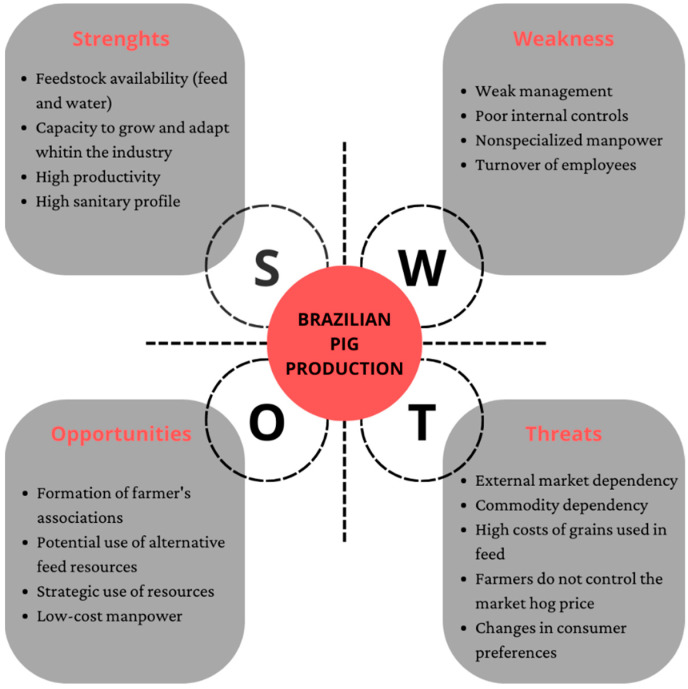
SWOT analysis of Brazilian pig production. Source: Research data.

**Table 1 animals-12-02229-t001:** Scheme adopted for allocation of swine production costs.

A—Variable costs
*I—Animal Costing Expenses*
1. Feed
2. Veterinary expenses
3. Reproductive management expenses
4. Consumer goods expenses
Subtotal animal costing
5. Transport, loading and insurance expenses
*II—Another Variable Expenses*
6. Taxes (ICMS, and others)
7. Guide of Animals Transit (GTA)
8. Funrural
9. Another variable taxes and fees
B—Fixed operating costs
*III—Manpower*
1. Fixed employees
2. Temporary workers
*IV—Telephony, Internet, Energy and Fuel*
3. Telephone and internet service expenses
4. Electric energy expenses
5. Fuel expenses
*V—Depreciations*
6. Housing
7. Machinery and implements
8. Biological assets
*VI—Maintenance and Conservation*
9. Housing maintenance
10. Machinery and implements maintenance
*VII—Another Fixed Expenses*
11. Taxes (ITR, and others)
12. Fees (Syndicate, Association, and others)
13. Another fixed taxes and fees
C—Operating cost (A + B)
D—Capital and land remuneration cost
*VIII—Opportunity Cost of Capital and Land*
1. Remuneration on fixed capital
2. Remuneration on working capital
3. Opportunity cost of land lease
E—Total cost (C + D)

Source: Research data.

**Table 2 animals-12-02229-t002:** Equations that make up the variable costs that compose the cost calculation model for swine production.

Equation	Equation Number
$CCA_{cf}=Pa_{cf}\times Crs_{cf}$	(4)
$Crs_{cf}=Crun_{cf}\times N_{cf}$	(5)
$Crun_{cf}=Ta_{cf}\times Qa_{cf}$	(6)
$CTA=\sum^{}CCA_{cf}$	(7)
$CS_{cf}=Pd_{cf}\times Qds_{cf}$	(8)
$Qds_{cf}=q_{cf}\times N_{cf}$	(9)
$CTS=\sum^{}CS_{cf}$	(10)
$CR_{f}=Pr_{cf}\times Qrs_{cf}$	(11)
$Qrs_{cf}=q_{cf}\times N_{cf}$	(12)
$CTR=\sum^{}CR_{cf}$	(13)
$CB_{i}=\frac{Pb_{i}\times Qb_{i}}{\omega }$	(14)
$CTB=\sum^{}CB_{i}$	(15)
CTC=CTA+CTS+CTR+CTB	(16)
$Cta_{l}=Pt_{l}\times Qt_{l}$	(17)
$Cseg_{l}=Txseg_{l}\times Cminvest_{l}$	(18)
$Cminvest_{l}=Nt_{l}\times \frac{kg}{cab}\times Pvc_{l}$	(19)
$CTtaseg=Ct_{l}+Cseg_{l}$	(20)
$CTdfv=\frac{ICMS+GTA+FUN+Otv}{\varphi }$	(21)
CV=CTC+CTtaseg+CTdfv	(22)

Source: Research data.

**Table 3 animals-12-02229-t003:** Acronyms and definitions of formulas used to calculate the variable costs that compose the calculation model for swine production.

Acronym	Definition
$CCA_{cf}$	Feeding cost, in Brazilian reais, for animal category *c* in phase *f*
$Pa_{cf}$	Price of the diet provided, in Brazilian reais per kg, for animal category *c* in phase *f*
$Crs_{cf}$	Weekly feed intake, in kg, for animal category *c* in phase *f*
$Crun_{cf}$	Amount of feed consumed, in kg, per animal of animal category *c* in phase *f*
$N_{cf}$	Number of animals from animal category *c* in phase *f*
$Ta_{cf}$	Supply period, in days per cycle, of feed for animal category *c* in phase *f*
$Qa_{cf}$	Amount supplied, in kg per day, of feed for animal category *c* in phase *f*
CTA	Subtotal of weekly food costs for the entire productive herd of the farm
$CS_{cf}$	Health cost, which includes vaccines and medicines, from animal category *c* in phase *f*
$Pd_{cf}$	Price of vaccine or medicine dose used in animal category *c* in phase *f*
$Qds_{cf}$	Number of vaccine or medicine doses used weekly in the animal category *c* in the phase *f*
$q_{cf}$	Number of doses used per animal of category *c* in phase *f*
CTS	Subtotal of weekly costs with health of the entire productive herd of the farm
$CR_{f}$	Cost with reproductive management, in Brazilian reais, in phase *f*
$Pr_{cf}$	Unit price of the input related to reproductive management consumed by animal category *c* in phase *f*
$Qrs_{cf}$	Weekly used amount of input consumed by animal category *c* in phase *f*
$q_{cf}$	Unit amount (or doses) consumed by animal category *c* in phase *f*
CTR	Subtotal of weekly costs with reproductive management
$CB_{i}$	Cost of consumer goods, in Brazilian reais, of different inputs *i*
$Pb_{i}$	Unit price of input *i*
$Qb_{i}$	Quantity consumed of input *i*
ω	Average number of weeks contained in a month, in a year with 365 days
CTB	Subtotal of weekly costs with consumables used by pig farming as a whole
CTC	Weekly subtotal of the cost of raising, in Brazilian reais
$Cta_{l}$	Weekly cost with transport of slaughter animals from batch *l*
$Pt_{l}$	Unit price, in Brazilian reais per kilometer, for the transport of slaughter animals from batch *l*
$Qt_{l}$	Number of kilometers traveled to transport slaughter animals from batch *l*
$Cseg_{l}$	Insurance cost, in Brazilian reais, of animals from lot *l*
$Txseg_{l}$	Fee charged to insure the batch of slaughter animals *l*
$Cminvest_{l}$	Average capital invested in lot *l*
$Nt_{l}$	Number of finished animals at the end of lot *l* housing
$Pvc_{l}$	Sale price, in Brazilian reais per live kilo, of the finishing pig sold in lot *l*
CTtaseg	Weekly subtotal for the cost of transport and insurance of animals in lot *l*
CTdfv	Weekly subtotal of costs with variable financial expenses, in Brazilian reais
ICMS	Tax on Transactions relating to the Circulation of Goods
GTA	Cost of issuing the Animal Transport Guide
FUN	Cost with the Rural Worker Assistance Fund
Otv	Cost of other expenses and variable fees
$CV_{l}$	Variable costs related to lot *l*

Source: Research data.

**Table 4 animals-12-02229-t004:** Equations that make up the fixed operating costs that compose the cost calculation model for swine production.

Equation	Equation Number
$CMO_{a}=\frac{S_{a}}{\omega }\times Nc_{a}$	(23)
$CTMO=\sum^{}CMO_{a}$	(24)
$CTE=\frac{Pe\times Qe}{\omega }$	(25)
$CTTI=\frac{Pt+Pi}{\omega }$	(26)
$CC_{x}=Pc_{x}\times Qc_{x}$	(27)
$CTG=\frac{\sum^{}CC_{x}}{\omega }$	(28)
$CDep_{j}=\frac{Vc_{j}-Vres_{j}}{vu_{j}}$	(29)
$Vc_{j}=Pc_{j}\times Qc_{j}$	(30)
$Vres_{j}=Vc_{j}\times txres_{j}$	(31)
$CTDep=\sum^{}\frac{CDep_{j}}{\varphi }$	(32)
$Cman_{j}=Vc_{j}\times txman_{j}$	(33)
$CTMan=\sum^{}\frac{CMan_{j}}{\varphi }$	(34)
$CTdff=\frac{ITR+Txf+Otf}{\varphi }$	(35)
$CFOP_{l}=CTMO+CTE+CTTI+CTG+CTDep+CTMan+CTdff$	(36)

Source: Research data.

**Table 5 animals-12-02229-t005:** Acronyms and definitions of formulas used to calculate the fixed operating costs that compose the calculation model for swine production.

Acronym	Definition
$CMO_{a}$	Weekly cost of labor, in Brazilian reais, by type of activity performed a
$S_{a}$	Employee salary with charges, in Brazilian reais per month, by type of activity performed a
$Nc_{a}$	Number of employees performing the activity a
CTMO	Subtotal of weekly labor costs on the farm
CTE	Weekly costs of electricity consumed by swine production
Pe	KWh price considering rural fare consumed
Qe	Monthly consumption of electricity by the swine production activity
CTTI	Weekly costs with telephone and internet
Pt	Price, in Brazilian reais, of the monthly telephone package used by the farm
Pi	Price, in Brazilian reais, of the monthly internet package used by the farm
$CC_{x}$	Monthly fuel cost, in Brazilian reais, by type of fuel x used
$Pc_{x}$	Price, in Brazilian reais per liter, of the type of fuel x consumed
$Qc_{x}$	Quantity, in liters, consumed monthly of each type of fuel x
CTG	Subtotal of weekly fuel consumption costs
$CDep_{j}$	Annual depreciation cost of capital asset j
$Vc_{j}$	Value, in Brazilian reais, of the capital asset j
$Pc_{j}$	Unit price, in Brazilian reais, of the capital good j
$\;Qc_{j}$	Unit quantity of fixed capital j
$Vres_{j}$	Residual value, in Brazilian reais, of capital asset j at the end of its useful life
$txres_{j}$	Fee for obtaining the residual value of fixed capital at the end of its useful life
CTDep	Subtotal of weekly costs with depreciation of swine production assets
$Cman_{j}$	Annual maintenance cost of the capital asset j
$\;txman_{j}$	Annual maintenance fee of capital asset j, in percentage
CTMan	Subtotal of weekly maintenance costs for machines, implements and facilities
CTdff	Weekly subtotal of costs with fixed financial expenses, in Brazilian reais
ITR	Rural land tax
Txf	Fixed annual fees, such as memberships in associations, unions, among others
Otf	Other fixed rates
$CFOP_{l}$	Fixed weekly operating costs

Source: Research data.

**Table 6 animals-12-02229-t006:** Equations that make up the opportunity costs of capital and land that compose the cost calculation model for swine production.

Equation	Equation Number
$RCI_{l}=\frac{trci_{j}\times \sum_{j}Vc_{j}}{\varphi }$	(38)
$RCG_{l}=\frac{trcg_{l}\times CTC}{\varphi }$	(39)
$RCT_{l}=\frac{ar_{r}\times PAT_{r}}{\varphi }$	(40)

Source: Research data.

**Table 7 animals-12-02229-t007:** Acronyms and definitions of formulas used to calculate the opportunity costs of capital and land that compose the calculation model for swine production.

Acronym	Definition
$RCI_{l}$	Remuneration of all fixed capital present in the activity necessary to produce a weekly batch *l*
$trci_{l}$	Fixed capital remuneration rate, in percentage per year
$RCG_{l}$	Remuneration of working capital, used to purchase inputs, in Brazilian reais per week
$trcg_{l}$	Working capital remuneration rate, in percentage per year
RCT	Remuneration for land use, in region *r* in period t, in Brazilian reais per week
$ar_{r}$	Area used by swine farming in region *r*, in hectares
$PAT_{r}$	Land lease price in region *r*, in Brazilian reais per hectare per year

Source: Research data.

**Table 8 animals-12-02229-t008:** Equations that make up the total cost that compose the cost calculation model for swine production.

Equation	Equation Number
$CTcab_{l}=\frac{CT_{l}}{Nt_{l}}$	(41)
$CT{@}_{l}=\frac{CT_{l}}{\left(Nt_{l}\times \frac{kg}{cab}\right)/@}$	(42)
$CTkg_{l}=\frac{CT_{l}}{Nt_{l}\times \frac{kg}{cab}}$	(43)

Source: Research data.

**Table 9 animals-12-02229-t009:** Acronyms and definitions of formulas used to calculate the total cost that compose the calculation model for swine production.

Acronym	Definition
$CTcab_{l}$	Cost, in Brazilian reais per head, of pig finished in lot *l*
$Nt_{l}$	Number of finished pigs at the end of lot *l* housing
$CT{@}_{l}$	Cost, in Brazilian reais per arroba, of animal finished in lot *l*
@	An arroba of swine produced which represents the unit value 18.75 kg
$CTkg_{l}$	Cost, in Brazilian reais per head, of pig finished in lot *l*

Source: Research data.

**Table 10 animals-12-02229-t010:** Equations that make up the economic indicators that compose the cost calculation model for swine production.

Equation	Equation Number
$Rs_{l}=\theta_{l}\times Pvc_{l}+\sum^{}\vartheta_{l}\times Pvd_{l}$	(48)
$Rcab_{l}=\frac{Rs_{l}}{Nt_{l}}$	(49)
$Rkg_{l}=\frac{Rs_{l}}{\theta_{l}}$	(50)
$L_{l}=Rs_{l}-CT_{l}$	(51)
$PN_{l}=\frac{CT_{l}}{Pvc_{l}\times Pcab_{l}}$	(52)
$RBC_{l}=\frac{Rs_{l}}{CT_{l}}$	(53)
$ROI_{l}=\frac{L_{l}}{CT_{l}}$	(54)
$RTs_{l}=L_{t}+CO_{l}+\beta$	(55)

Source: Research data.

**Table 11 animals-12-02229-t011:** Acronyms and definitions of formulas used to calculate the economic indicators that compose the calculation model for swine production.

Acronym	Definition
$Rs_{l}$	Total weekly revenue, in Brazilian reais, from the weekly sale of market pigs and culling animals
$Pvc_{t}$	Selling price, in Brazilian reais per live kilo, of the market pig sold in lot *l*
$Pvd_{l}$	Sale price, in Brazilian reais per live kilo, of the cull animal sold in lot *l*
$Rcab_{l}$	Average weekly revenue, in Brazilian reais per head of market pig produced in batch *l*
$Rkg_{l}$	Average weekly revenue, in Brazilian reais per kilogram of market pig produced in lot *l*
$L_{l}$	Economic profit related to the production and marketing of the batch *l* of animals
$RTkg_{l}$	Total weekly income to the producer, in Brazilian reais per kilograms produced in batch *l*
$PN_{l}$	Leveling point for batch *l*, in number of market pigs
$RBC_{l}$	Benefit/cost ratio, in Brazilian reais, for lot *l* under analysis
$ROI_{l}$	Return on investment, in percentage, for lot *l* under analysis
$RTs_{l}$	Total weekly income to the producer, per batch *l* produced
β	Remuneration, in Brazilian reais per week, for the work of the producer and his family

Source: Research data.

## Data Availability

The data presented in this study are openly available in *Teses USP* repository, at https://doi.org/10.11606/D.10.2021.tde-19072021-122134.

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
