# Peer review of "Development of a Swine Production Cost Calculation Model"

_animals, 2022, doi:10.3390/ani12172229_

Round 1

Reviewer 1 Report

Dear Authors

This is a very complex work on the use of predictive models.

I think that the simple summary should be improved. That summary should be addressed to people who have no knowledge about the topic. However, it does not give accessible information.

The introduction should be improved. Current citations are needed, and compare the possibility of applying this model in other markets worldwide.

I am not qualified to judge the results.

More current citations should be added to the discussion and discussed.

Thank you

Author Response

Dear Reviewer 1, we appreciate your comments and compliments to this manuscript. Please check the authors’ answers to your concerns below.

Comments to Author: 

“I think that the simple summary should be improved. That summary should be addressed to people who have no knowledge about the topic. However, it does not give accessible information…”

ANSWER: Dear Reviewer 1, the simple summary was changed.

Comments to Author: 

“The introduction should be improved. Current citations are needed, and compare the possibility of applying this model in other markets worldwide…”

ANSWER: Dear Reviewer 1, changes were made on the introduction section, also new relevant papers were included in the text.

Comments to Author: 

“More current citations should be added to the discussion and discussed…”

ANSWER: Dear Reviewer 1, some pertinent current articles were added in the discussion section.

Reviewer 2 Report

Dear Authors, 

thank you for this valuable piece of work, which I find both interesting and necessary. Indeed, I agree with you about the need to provided farmers with tools to improve profitability meanwhile trying to stay on the market and face competition. So, despite the approach could be intuitive and reasonable, I do believe that the need of such tools (like the model you deveolped) may turn useful to primary production into facts, that is to say  be put into practice. 

This being said, I also believe that you can improve your paper with some small adjustments.

I would carefully use the terms like pork production which is different from pig farming. So, not pig production but pork production (it is the end  product further processed or not, which sets the final price on the market, I mean the slaughtered pig). In this way, you don't mix the different phases of the production chain.

The paper is well developed, however I believe that it must be also considered what happens worldwide in the pig sector, prior to report the Brazilian case. I mean to say that intensive pork production is now facing the revolution of the PLF and brand protection, thereby increasing the consumers awareness who drive the demand. The same, welfare friendly productions also affect in way the whole value chain is going to be conceived. I believe that an overview of the global context is necessary. I would suggest you to refer to Cappai et al. 2018, Computers and Electronics in Agriculture,145, Pages 248 - 252 doi:10.1016/j.compag.2018.01.003 for an idea.

I would then suggest to start with a SWOT analysis to depict the state of the art of Brazilian procution, maybe directly reported into a table with strength points, weaknesses, opportunities and Threats.

Last but not least, in M&M, in the paragraphs 2.1 and 2.2 please, be more structured, rather wordy.

Thank you.

Author Response

Dear Reviewer 2, we appreciate your comments and compliments to this manuscript. Please check the authors’ answers to your concerns below.

Comments to Author: 

“I would carefully use the terms like pork production which is different from pig farming. So, not pig production but pork production (it is the end  product further processed or not, which sets the final price on the market, I mean the slaughtered pig). In this way, you don't mix the different phases of the production chain.…”

ANSWER: Dear Reviewer 2, the changes were made.

Comments to Author:

“The paper is well developed, however I believe that it must be also considered what happens worldwide in the pig sector, prior to report the Brazilian case. I mean to say that intensive pork production is now facing the revolution of the PLF and brand protection, thereby increasing the consumers awareness who drive the demand. The same, welfare friendly productions also affect in way the whole value chain is going to be conceived. I believe that an overview of the global context is necessary. I would suggest you to refer to Cappai et al. 2018, Computers and Electronics in Agriculture,145, Pages 248 - 252 doi:10.1016/j.compag.2018.01.003 for an idea…”

ANSWER: Dear Reviewer 2, the changes were made. Also, the Cappai et al. (2018) article information was added on the discussion section.

Comments to Author:

“I would then suggest to start with a SWOT analysis to depict the state of the art of Brazilian procution, maybe directly reported into a table with strength points, weaknesses, opportunities and Threats…”

ANSWER: Dear Reviewer 2, the SWOT analysis was developed and included in the article.  

Comments to Author:

“Last but not least, in M&M, in the paragraphs 2.1 and 2.2 please, be more structured, rather wordy…”

ANSWER: Dear Reviewer 2, the changes has been made, and the paragraphs has been shortened.